

# Thermobaric circulation in a deep freshwater lake

Joshua Marks[1, 2], Kazuhisa A. Chikita[3], and Bertram Boehrer[1, 2]

[1]Helmholtz Centre for Environmental Research - UFZ, Brueckstrasse 3a, 39114 Magdeburg, Germany
[2]Department of Physics and Astronomy, Heidelberg University, Im Neuenheimer Feld 226, 69120 Heidelberg, Germany
[3]Arctic Research Center, Hokkaido University, Sapporo 001-0021, Japan

**Correspondence:** Joshua Marks (joshua.marks@ufz.de)

**Abstract.** Numerical lake models are a powerful tool to optimize water management and mitigate changes due to climate change. Hence, detailed implementation of lake specific processes is crucial to ensure optimal results. However, common numerical lake models have so far omitted the effect of thermobaricity despite its significant influence on deep water circulation in deep lakes. The thermobaric effect is based on the temperature dependence of the compressibility of water. As a conse-
quence, deep water can be significantly colder than $4\,^\circ\text{C}$ and deep water renewal becomes complex. For a proper investigation, numerical models can be appropriate tools to display and understand such processes better. Inspired by Lake Shikotsu, which is an excellent example for the influence of thermobaricity, we developed a simplified 1D model for thermobaric effects. Here, we used in situ density to replace potential density for stability considerations such as the Brunt-Väisälä frequency. To prevent any competing influences and isolate thermobaric effects, we excluded any external forcing except for the surface tempera-
ture input. Accordingly, we excluded salinity, chose a cylindrical bathymetry without shallow areas, and omitted any inflows. Therefore, the model reproduced deep water circulation solely based on thermal forcing at the surface. We were able to identify key features of the deep water renewal events as well as different phases of the mixing period. Additionally, we investigated the influence of previous deep water renewal events and the current surface temperature on the deep water circulation. Our results emphasize the feasibility and necessity of the implementation of thermobaricity in numerical lake models.

## 1 Introduction

Across all climate zones, lakes and reservoirs are undergoing changes due to climate change (Adrian et al., 2009; Sun et al., 2025) and heavy human impact (Søndergaard and Jeppesen, 2007; Weyhenmeyer et al., 2024). These water bodies provide central services such as water and food supply, energy production, flood prevention, and space for recreation. In particular, the provision of high quality drinking water is becoming an increasing challenge that humanity has to deal with (Delpla et al.,
2009). This accounts globally and awareness is growing. Hence, to ensure optimal management of lakes and reservoirs in the future, proper knowledge of processes and the prediction of changes are crucial. Numerical modeling of lakes serves as a central tool to predict future development and optimize management strategies (e.g. Weber et al., 2019; Mi et al., 2022) to restore and maintain a healthy ecological state of our natural environment. Good management can help adapt our strategies and mitigate the influences of climate change and direct human impact at least in part (e.g. Winton et al., 2019; Regev et al., 2025).
25       Numerical models for lakes are often based on assumptions originating from oceanography. As a consequence, the inclusion



of lake specific properties of limnic waters can improve simulations of lakes fundamentally. This spans from using local weather conditions to implementing lake specific solute compositions in density functions (Pawlowicz and Feistel, 2012; Moreira et al., 2016). In this paper, we want to demonstrate the necessity of including the temperature dependence of the compressibility in the water properties. The effect deriving from this property is called thermobaric effect (McDougall, 1987). Thermobaricity

has so far not been dealt with in the commonly used lake models. However, some ocean models changed to in situ density when they changed to the new ocean standard TEOS-10 (IOC et al., 2010), which includes the effect.

The effect of thermobaricity is based on the temperature dependence of the compressibility of water (which is tantamount to the pressure dependence of the thermal expansion of water, $\partial^2 \rho_{\mathrm{in-situ}}/(\partial\theta\partial p)$) (McDougall, 1987). For pure water, this leads to a decrease of the temperature of maximum density $T_{\mathrm{md}}$ from approximately $3.98\,^{\circ}\mathrm{C}$ at atmospheric pressure by about

$0.02\,^{\circ}\mathrm{C}\,\mathrm{bar}^{-1}$ (Chen and Millero, 1977). For instance, pure water at the pressure of $500\,\mathrm{m}$ depth has its temperature of maximum density at about $3\,^{\circ}\mathrm{C}$. Ultimately, this effect allows water with temperatures below $3.98\,^{\circ}\mathrm{C}$ to be denser than slightly warmer water below a certain depth. As a consequence, we find deep water significantly colder than $3.98\,^{\circ}\mathrm{C}$ in lakes of the temperate climate zone (e.g. Carmack and Weiss, 1991; Crawford and Collier, 1997; Boehrer et al., 2009, 2013).

Also connected to the temperature of maximum density, the process of cabbeling differs from thermobaricity (McDougall,

1987; Carmack and Weiss, 1991; Grace et al., 2023a, b). Cabbeling originates from thermal bars, where mixing of two water parcels across the temperature of maximum density results in even denser water, which itself drives convective circulation (cabbeling) (Ivey and Hamblin, 1989; Shimaraev et al., 1993). Although deep water renewal in some lakes is controlled only by thermobaricity, also cabbeling may be involved in the deep mixing and deep water formation. The vertical progression of cold surface water to the abyss is complex and the depiction of density differences becomes difficult as the convenient property

of potential density is lost when thermobaricity is dealt with. External forcing by strong winds as driving force for the deep water renewal has been discussed (e.g. Weiss et al., 1991; Boehrer et al., 2013). Here, these winds push cold water beyond the compensation depth and from there it can proceed due to its higher in situ density compared to the surrounding water. Killworth et al. (1996) and Piccolroaz and Toffolon (2013) created models to describe such deep water renewal in Lake Baikal. They used wind as external forcing to represent downwelling under conditions of thermobaricity.

In contrast to them, we simulated the deep water mixing controlled by thermobaricity instead of forced plume downwelling. Therefore, we decided to keep everything at the minimum complexity to depict the thermobaric effects as well isolated as possible. Consequently, we considered a cylindrical water domain without lateral gradients (on the considered length scale) for our 1D model. We prohibited shallow areas for the formation of waters of different properties, valleys to guide submerged flows, and horizontal gradients at the surface. Additionally, we implemented pure water properties (no salinity). Lake Shikotsu,

Hokkaido, Japan, served as our inspiration, because thermobaric stratification has been documented there (Boehrer et al., 2008, 2009). Hence, we chose the temperature range and size of the basin accordingly, hoping that our numerical 1D model would manage to reproduce water circulation patterns and temperature profiles with some similarities to Lake Shikotsu. As we had presumed that the deep water circulation in this lake could be understood as a one dimensional circulation, we believed that our parsimonious model should represent the fundamental features there. Nevertheless, we expected not too close

similarity, since we abstained from including any detailed information of bathymetry or forcing. Admittedly, Lake Shikotsu is





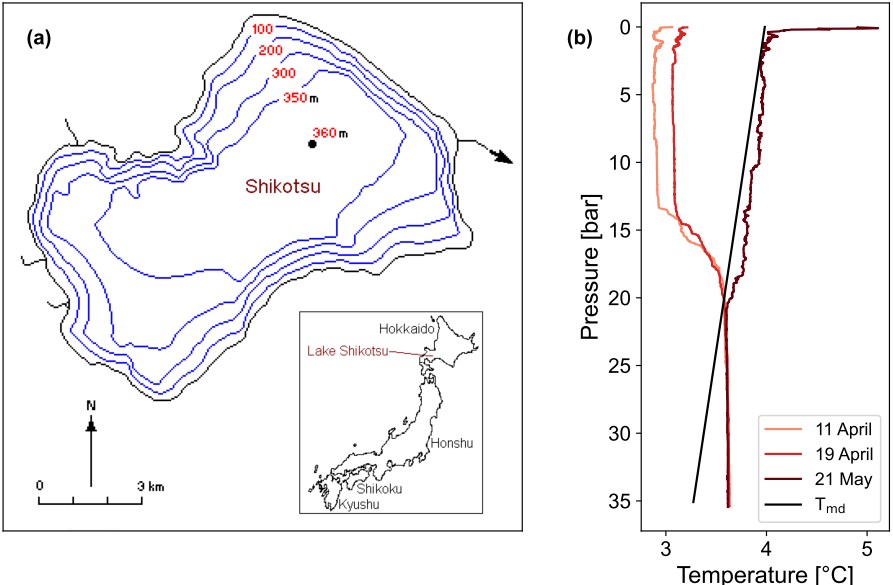

**Figure 1.** Location, bathymetry and temperature profiles from 2005 of Lake Shikotsu. **(a)** presents the bathymetry map of Lake Shikotsu and its location in Japan (adapted from Boehrer et al., 2009), while **(b)** shows temperature profiles from 2005 (data from Boehrer et al., 2008) and the temperature of maximum density $T_{\mathrm{md}}$.

a particularly beautiful representation of thermobaricity, but we hope to convince the readers of the importance of including temperature dependent compressibility (thermobaricity) in numerical models, if they want to have a realistic simulation of deep water movements in lakes with deep water temperatures close to the temperature of maximum density.

## 2 Methods

### 2.1 Inspiration


Our 1D model was inspired by Lake Shikotsu, Hokkaido, Japan. This lake is located in the temperate zone and consistently experiences temperatures below $3.98\ °C$ during winter. It is a volcanic caldera lake of collapse type, formed by the great eruption of Shikotsu Volcano about $40,000$ years ago. Because of that, the lake has a roughly cylindrical shape, very steep sidewalls, and a maximum depth of $360\ \mathrm{m}$ (Fig. 1a). Boehrer et al. (2008, 2009) investigated the deep water circulation and its

dependency on thermobaricity. The temperature profiles, measured in 2005, clarify this dependency (Fig. 1b). They cross the $T_{\mathrm{md}}$ line during the winter stratification and resemble it above this intersection shortly after. At the same time, the deep water temperature below the intersection stays isothermal throughout. The very steep sidewalls and comparatively small horizontal extend of the lake, as well as its symmetric shape, suggest a rather one dimensional governance of the system. Missing shallow





areas along the shore and the uniform slope without any trenches make the accumulation and downwelling of cold water plumes
induced my external forcing unlikely.

## 2.2   Density and Stability

Potential density cannot include the pressure dependency of the density and therefore neglects thermobaric effects. To proper represent thermobaricity in a model, the use of in situ density $\rho_{\mathrm{in-situ}}$ instead of potential density $\rho_{\mathrm{pot}}$ is necessary. To start with, the sound velocity $c$ is directly connected with the compressibility of water $\beta_{\mathrm{S}}$ by

$$\beta_{\mathrm{S}} = \frac{1}{\rho_{\mathrm{in-situ}}} \left( \frac{\partial \rho_{\mathrm{in-situ}}}{\partial p} \right)_{\mathrm{S}} = \frac{1}{\rho_{\mathrm{in-situ}} \cdot c^2} \tag{1}$$

(Meschede, 2015), where $p$ is the pressure and the index S indicates an adiabatic process, and therefore represents the temperature dependency of the compressibility (thermobaricity). Hence, we used this to obtain a formulation of the in situ density. According to Eq. (1), the sound velocity can be expressed by

$$\frac{1}{c^2} = \left( \frac{\partial \rho_{\mathrm{in-situ}}}{\partial p} \right)_{\mathrm{S}}. \tag{2}$$

From Eq. (2) the in situ density can be derived as

$$\rho_{\mathrm{in-situ}} = \rho_{\mathrm{pot}} + \int\limits_0^p \frac{1}{c^2}\,\mathrm{d}p. \tag{3}$$

Here, in situ density and sound velocity depend on temperature, salinity, and pressure while potential density only depends on temperature and salinity. In our approach, we exclude salinity to prevent distraction by competing effects. Hence, salinity is disregarded in the following and we use the formulations of Tanaka et al. (2001) for potential density and Belogol'skii et al.
(1999) for sound velocity. We are fully aware that for limnic water the solutes must be included in the water properties for the calculation of density (Boehrer et al., 2010; Moreira et al., 2016) and sound velocity when salinity is considered.

Equation (3) is then further simplified by applying a linear approximation for $\frac{1}{c^2}$ with respect to pressure

$$\frac{1}{c^2(p)} = mp + n, \text{ with } m = \frac{\partial \frac{1}{c^2(p)}}{\partial p} \text{ and } n = \frac{1}{c^2(p=0)}. \tag{4}$$

For the modeled depth range up to 360 m (compare Sect. 2.3) and the interesting temperature range close to the temperature
of maximum density $T_{\mathrm{md}}$, roughly between 3 and 4 °C, a maximum deviation of $2.13 \cdot 10^{-12}$ s$^2$ m$^{-2}$ of the sound velocity is introduced. This deviation is small relative to the change of $\frac{1}{c^2}$ with respect to temperature or pressure, which are in the order of $10^{-9}$ s$^2$ m$^{-2}$.

The final formulation of the in situ density is

$$\rho_{\mathrm{in-situ}}(T,p) = \rho_{\mathrm{pot}}(T) + \frac{p^2}{2} \frac{\partial \frac{1}{c^2(T,p)}}{\partial p} + \frac{p}{c^2(T,p=0)}. \tag{5}$$

While we clearly distinguish between $\rho_{\mathrm{in-situ}}(T,p)$ and $\rho_{\mathrm{pot}}(T)$, we only use potential temperature $T$ (and never in situ temperature $T_{\mathrm{in-situ}}$; differences would be in the range of 1 mK in the presented case).





With the explicit formulation of the in situ density it is now possible to compare neighboring water parcels at the same pressure to check for density differences. Also, the Brunt-Väisälä frequency, which uses the density gradient without the pressure influence for stability considerations, can be calculated straight forward by using the in situ density at the same
pressure as

$$N^2 = -\frac{g}{\rho_{\text{in-situ}}} \cdot \left( \frac{\partial \rho_{\text{in-situ}}}{\partial z} - \frac{\partial \rho_{in-situ}}{\partial p} \frac{\partial p}{\partial z} \right) = -\frac{g}{\rho_{\text{in-situ}}} \frac{\rho_{\text{in-situ}}(T_2, p_2) - \rho_{\text{in-situ}}(T_1, p_2)}{z_2 - z_1}, \quad (6)$$

where $z$ is the depth and the indices indicating the quantity of the corresponding water parcel.

## 2.3 Numerical Model

We created a simple 1D model to conceptually simulate deep water renewal based on thermobaricity. The model domain con-
sisted of 180 layers of $2\,\text{m}$ each, which resulted in a maximum depth of $360\,\text{m}$. The values were motivated by Lake Shikotsu because this kind of deep water renewal is suspected to have a significant influence in this lake. The lateral extent of the layers was not specified, but the volume and hence the horizoantal extent of all layers was considered equal. Lateral gradients were excluded by the one dimensional design of the model domain (Boehrer et al., 2008).

Our intention was the demonstration of thermobaricity. Hence, in contrast to common lake models, we removed the atmo-
spheric forcing at the lake surface. Instead, we used surface water temperature as a boundary condition (Boehrer et al., 2000). As a consequence, the numerical model was exclusively forced by temperature, and resulting density differences, in the surface layer of the model. The surface temperature input is also based on the measured surface temperature from Lake Shikotsu from 18 October 2023 to 8 Mai 2024, which includes temperatures significantly colder than $4\,°\text{C}$ during the winter (compare Fig. 2). We decided to use real surface data to keep heat exchange with the atmosphere in a realistic range. The surface temperature
series was then linearly extrapolated to a duration of one year. Although surface temperatures during the summer are unrealistically represented by the linear extrapolation, the thermobaric effect is unaffected by temperatures above $4\,°\text{C}$ and the summer temperatures are much warmer. The annual surface temperature is then repeated over seven years which is the total length of the simulation. To include the effect of day-night variation, the time step was set to $1\,\text{h}$.

Starting from an arbitrary isothermal temperature profile, the temperature of the uppermost layer was inserted. We realized
the diffusion by exchanging half of the volume of each layer equally with both neighboring layers and homogenizing each layer correspondingly. By doing so, the time step size and layer size influence the amount of diffusion. For the values used, the diffusion is about $5.5 \cdot 10^{-4}\,\text{m}^2\,\text{s}^{-1}$, which is roughly in the expected order of magnitude (e.g. Saber et al., 2018).

The special feature of our model is the comparison of in situ densities at given pressure, instead of using potential densities, as basis for the stability, and therefore mixing, of the water column. This means that in situ densities of two neighbouring layers
are compared at the pressure of the lower one, while the temperatures (and in principle also the salinity) are kept the same for the two layers, respectively. If the stratification is stable, there is no action. However, if water in the lower layer is less dense than in the upper, the layers are mixed and the temperatures of the two layers are averaged. This comparison is then iteratively done for all layers in the water column. If the instable (mixed) layers are connected to the uppermost (controlled) layer, the temperatures of all layers are set to the prescribed surface temperature.





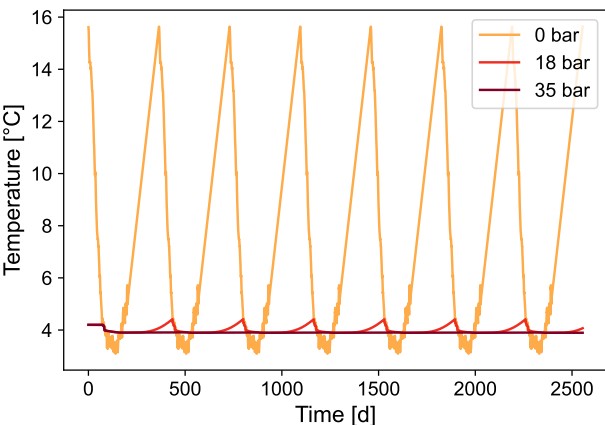

**Figure 2.** Temperature logs at different depths. The logs show the temperature progression of the controlled surface temperature (0 bar) and at depths of 18 and 35 bar over the entire simulation period.

To demonstrate that thermobaric conditions form from an isothermal profile of $4.2\ ^{\circ}\text{C}$ and remain thereafter, we ran the model for seven years. Hence, we can use later years when thermobaric conditions have established to discuss the resulting circulation features. In addition, two more runs were done by changing the input surface temperature in the fourth year by adding and subtracting $0.4\ ^{\circ}\text{C}$, respectively. They provide an impression of the stability of thermobaricity and the effect of variable winters on deep water renewal.

## 3   Results

The model is driven by the input of the surface temperature. We follow the ocean convention of referring to hydrostatic pressure of $0\ \text{bar}$ at the water surface. The simulated temperatures at mid depth ($18\ \text{bar}$) and the bottom ($35\ \text{bar}$) are also displayed for the entire simulation time (Fig. 2). At mid depth, the temperature rises during the summer stratification and follows the surface temperature to a temperature slightly below $4\ ^{\circ}\text{C}$ during autumn and winter. This temperature is also equal to the temperature at $35\ \text{bar}$, where only little temperature variation can be observed during the simulation except for the first winter, because the starting temperature profile was set to $4.2\ ^{\circ}\text{C}$.

### 3.1   Temperature profiles

For the discussion of typical thermobaric circulation features, we refer to the fourth year of simulation, as we considered the first three years as possibly affected by the initial conditions. We divide the year into typical stratification phases: winter cooling (WC), winter stratification (WS), summer warming (SW) and summer stratification (SS). The order of these phases is oriented on the starting point of the simulation, which is in October, the beginning of the measured surface temperature input. Figure 3 exhibits four exemplary temperature profiles for each phase.



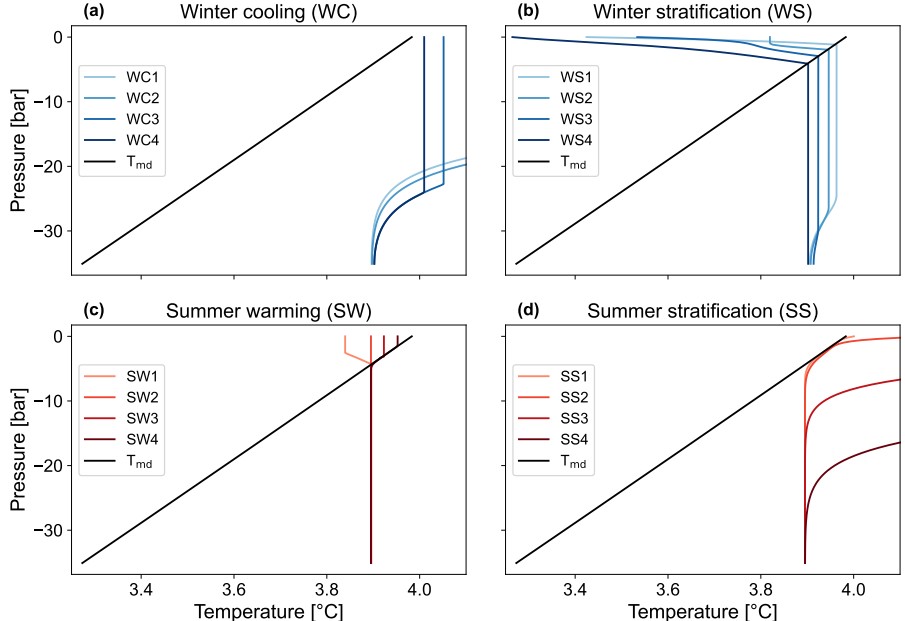

**Figure 3.** Temperature profiles of the fourth year of the simulation period. The selected temperature profiles are split into four plots for clarity. **(a)** contains profiles from the winter cooling (WC) phase, **(b)** from the winter stratification (WS) phase, **(c)** from the summer warming (SW) phase, and **(d)** from the summer stratification (SS) phase. Their points in time are marked in Fig. 4. Additionally, all plots include the temperature of maximum density $T_{\mathrm{md}}$.

The four exemplary profiles of the winter cooling (WC1: 12 November, 20:00; WC2: 29 November, 12:00; WC3: 7 January, 01:00; WC4: 7 January, 10:00) are shown in Fig. 3a. During this phase, the surface water cools rapidly and circulates the water below to a depth where colder temperatures still stabilize the water column. Additionally, due to the high surface temperatures compared with the deep water temperatures, the deep water is still subject to warming (compare WC1 and WC2 with WC3 and WC4). This behavior continues until the surface water reaches the temperature of maximum density $T_{\mathrm{md}}$ at the surface, which is approximately $3.98\,^{\circ}\mathrm{C}$. From that point on, the inverse winter stratification begins.

During the winter stratification, the surface water is further cooled and it stratifies inversely (compare profiles in Fig. 3b: WS1: 21 January, 14:00; WS2: 29 January, 07:00; WS3: 20 February, 14:00; WS4: 19 March, 16:00). This means that close to the surface colder water floats on warmer water due to its lower density. Inverse stratification is stable to the point where the temperature profile intersects the $T_{\mathrm{md}}$ line. Water mixes across $T_{\mathrm{md}}$ and produces denser water similar to cabbeling (Grace et al., 2023a, b). This denser water circulates the water column below due to thermobaricity to the depth where lower temperatures stabilize the water column. Since the surface water is further cooled in this phase, the intersection with $T_{\mathrm{md}}$ deepens and ever colder water mixes into the deep water. Eventually, the bottom layer is replaced by colder water from above and the circulation extends to the deepest point. Hence, the intersection with $T_{\mathrm{md}}$ controls the deep water temperature. Short intermediate warming events at the surface (e.g. WS2) induce small mixing cells at the surface, similar to the winter cooling, but



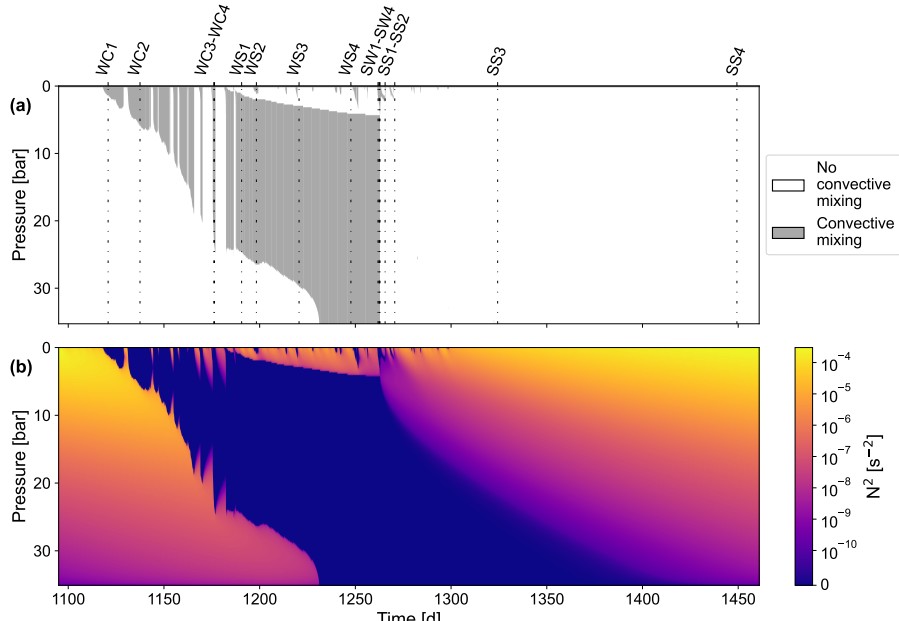

**Figure 4.** Convective mixing cells and stability of the fourth year of the simulation period. **(a)** shows the convective mixing cells and the points in time of the profiles shown in Fig. 3 are marked with dash-dotted lines. **(b)** depicts the corresponding Brunt-Väisälä frequency, calculated with Eq. (6).

do not influence the deep water mixing cells or their progression. The winter stratification comes to an end when the surface temperature starts to rise significantly.

During the summer warming (profiles in Fig. 3c: SW1: 2 April, 20:00; SW2: 3 April, 09:00; SW3: 3 April, 13:00; SW4: 3 April, 21:00) the inverse stratification gets erased by the rising surface temperatures. Since the surface layer temperature is below 3.98 °C, the warming surface water mixes the colder water below due to the higher density and we see a homogenized surface layer. From the moment when the surface layer temperature exceeds the deep water temperature (WS2), cabbeling comes to an end. The circulation of the deep water stops and a stable density stratification establishes along $T_{\mathrm{md}}$ above the

isothermal deep water (SW3 and SW4).

The following summer stratification (profiles in Fig. 3d: SS1: 6 April, 13:00; SS2: 11 April, 14:00; SS3: 4 June, 08:00; SS4: 7 October, 08:00) starts as soon as surface temperatures exceed 3.98 °C. Stable density stratification can establish from the surface. Early in this phase, the temperature profile follows the $T_{\mathrm{md}}$ line from the surface into the depth where $T_{\mathrm{md}}$ is equal to the uniform deep water temperature. However, during the summer period, diffusive transport from the surface increases the

temperature in ever deeper water top down.





## 3.2 Convective mixing

In autumn, cooling at the surface drives vertical convection (Fig. 4a) the same way as if thermobaricity was not included. The surface temperature is decreasing and the mixing extends vertically to the depth where the surface temperature is equal to the current temperature at this depth. This depth also decreases due to the colder surface temperatures (compare WC1-4 in Fig.
3a). The mixing is interrupted from short rewarming events but is always connected to the surface.

Before reaching the bottom, temperatures at the surface fall below $3.98\,°\mathrm{C}$ and the stable inverse stratification terminates the circulation at the surface. Hence, the convection cell detaches from the surface (Fig. 4a). As it is permanently overlain by colder temperatures, no interruptions happen due to small rewarming events. They only induce additional small mixing cells at the surface, which are well seperated from the deep circulation cell. Moreover, the deepening of the intersection of the temperature
profile with the $T_{\mathrm{md}}$ line (compare WS1-3 in Fig. 3b) leads to a deepening of the upper end of the deep circulation cell. Finally the mixing cell reaches the bottom (compare WS4 in Fig. 3b).

The convection in the deep water layers stops when the conditions for cabbeling disappear. This happens when surface temperatures rise above deep water temperatures (SW2 Fig. 3c). The two mixing cells, the mixing at the surface and the deep water circulation, remain well separated. Afterwards, the deep convection stops and only the surface mixing cell remains in
place (compare SS3 and SS4 in Fig. 3c), even though its depth decreases due to the change of $T_{\mathrm{md}}$ with depth. Below the $T_{\mathrm{md}}$ line, the warmer surface water is less dense than the colder deeper water, which limits the extent of the surface mixing cell.

After the surface temperature crosses $3.98\,°\mathrm{C}$, only short intermittent cooling events can induce small surface mixing cells. In general, the water column becomes more stable and no mixing occurs until the end of the yearly cycle in September (compare SS1-4 in Fig. 3d).

## 3.3 Stability

The display of convection cells (Fig. 4a) is instructive for understanding vertical transport, but a quantitative approach to stability is desirable. Potential density is not applicable and in situ density is dominated by compression and hence does not provide much insight into stability either. However, we showed that the stability frequency can be calculated (Eq. (6), Fig. 4b). The convection cells can be identified as areas of stability $0\,\mathrm{s}^{-2}$. They are sharply separated from areas of higher
stability. Starting in October, the water column becomes unstable from the surface due to decreasing surface temperatures. Short rewarming events partly restabilize the instable part at its upper and lower end. After the detachment of the mixing cell from the surface, a stable layer develops at the surface. Its lower end corresponds to the depth where the temperature profile intersects with the $T_{\mathrm{md}}$ line. The surface mixing cells originating from rewarming events can be identified by destabilized parts at the surface. After the deep convective mixing ceases and the summer stratification sets in, the whole water column stabilizes
due to the increasing surface temperatures. Besides that, a marginally stabilized part remains close to the bottom for a long period. Towards the end of the summer stratification, it also becomes increasingly stable.



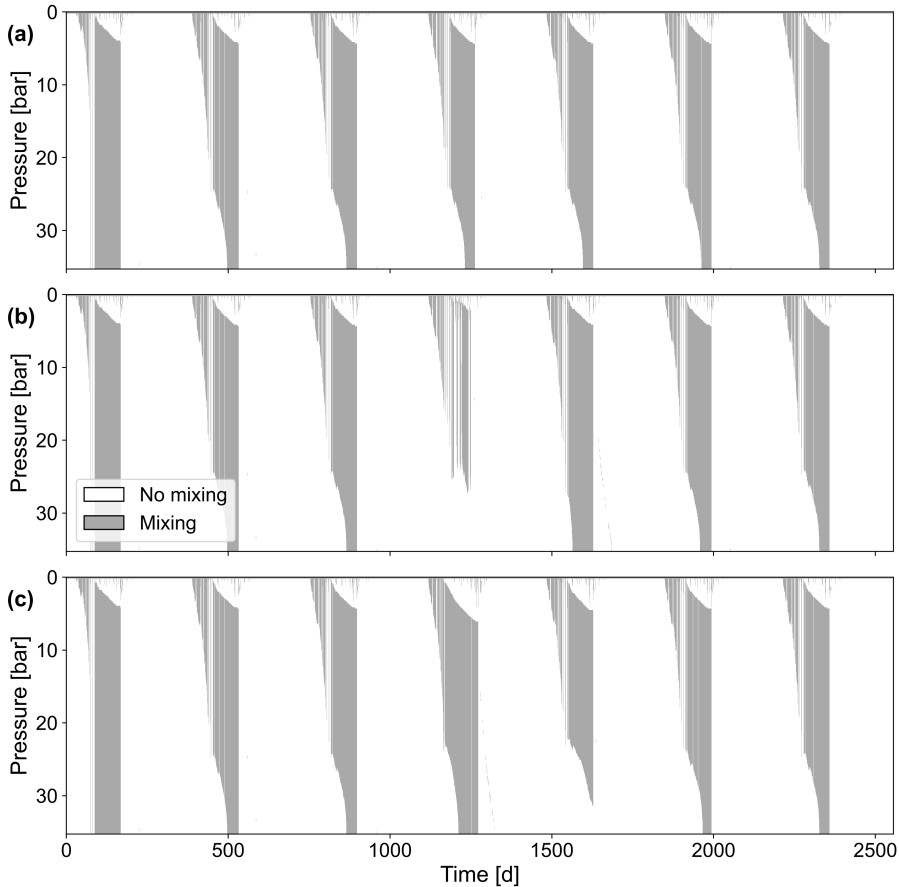

**Figure 5.** Mixing cells over the entire simulation period for three variants of surface temperature input. The surface temperature of the fourth year of the simulation period was **(a)** not modified, modified by **(b)** $+0.4\,^\circ$C, and **(c)** $-0.4\,^\circ$C.

## 3.4 Interannual Variations

To check the effect of interannual variability, we added two more runs by elevating or lowering the surface temperature of the fourth year of the simulation period by $0.4\,^\circ$C, respectively. Increasing the surface temperature leads to a weaker mixing during winter (Fig. 5b). A shorter and warmer cold period leads to less cold water at the surface. Hence, the crossing of the temperature profile with the $T_{\mathrm{md}}$ line deepens less and the deep water circulation is weakened. Depending on the strength of this change, mixing may not reach the bottom (compare Fig. 5b). Consequently, the temperature of the deep water is increased in the following winter and the convective mixing reaches the bottom earlier in that year.

In contrast, decreased winter temperatures enhance the mixing during winter (Fig. 5c). The colder temperatures and prolonged cold period leads to further deepening of the intersection of the temperature profile with $T_{\mathrm{md}}$. Therefore, colder water is mixed into the deep water and at the end of the deep circulation period colder isothermal deep water has been formed. As



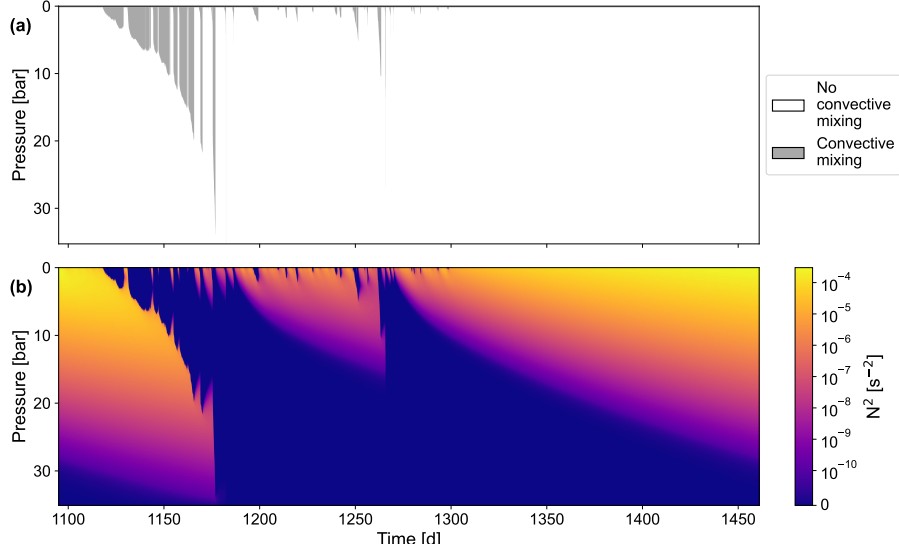

**Figure 6.** Convective mixing cells and stability of the fourth year of the simulation period for the simulation using potential density $\rho_{\mathrm{pot}}$ instead of in situ density $\rho_{\mathrm{in-situ}}$ (compare Fig. 4). **(a)** shows the convective mixing cells and **(b)** depicts the corresponding Brunt-Väisälä frequency, calculated with Eq. (6).

a result, the following year with the original surface temperature input has less deep mixing cells since the colder deep water impedes deep mixing to the bottom (compare Fig. 5c).

In general, the variations in the surface temperature of successive years are in interplay with each other. They can either
reinforce (e.g. a prior strong summer stratification and current strong winter period) or weaken (e.g. a prior weak summer stratification and current strong winter period) each other. Therefore, the model not only exhibits the different behavior of the mixing phases based on the variations in the forcing, the surface temperature, for the manipulated year, but also the dependency on the prior history. In general the stratification returns to normal after only few years.

### 3.5 Simulation with potential density

To demonstrate the effect of thermobaricity, we conducted the exact same simulation as in Sect. 2.3 only substituting in situ density $\rho_{\mathrm{in-situ}}$ with potential density $\rho_{\mathrm{pot}}$ (i.e. neglecting thermobaric effects). We used the formulation of Tanaka et al. (2001) for $\rho_{\mathrm{pot}}$ as before. Cabbeling was included as an instantaneous destabilizing effect by setting the surface temperature equal to $T_{\mathrm{md}}$ in one time step for every transition of maximum $\rho_{\mathrm{pot}}$. Neglecting thermobaric effects change the size and the period of convective mixing cells and the water column stability fundamentally (Figure 6). There is only one significant period
of overturn after the summer stratification culminating in a brief complete overturn when temperatures cross $T_{\mathrm{md}}$ and go into winter stratification. Another brief moment of deep overturn occurs when temperatures go into summer stratification, again



crossing $T_{\mathrm{md}}$. The small mixing cells at the surface still occur due to their origin from small rewarming events which are unaffected by the thermobaric effect.

## 4  Discussion

With our numerical approach we intended to demonstrate (a) the effect of thermobaricity on a simple one dimensional system and (b) the proper representation of thermobaricity by stability considerations based on in situ density. Key features of this idealized thermobaric system were (1) the isothermal deep water, (2) the crossing of the temperature profile with the temperature of maximum density $T_{\mathrm{md}}$ line and its control of the deep water temperature, (3) the period when surface water and deep water convection cells are separated, as well as (4) the alignment of the temperature profile with the $T_{\mathrm{md}}$ line in the upper water

shortly after the winter stratification.

Despite the simplicity of our numerical approach, the key features (1)-(4) are remarkably clearly visible in the temperature profiles of Lake Shikotsu in 2005 (Fig. 1). The basic agreement of these features show that these thermobaric features in Lake Shikotsu (and other similar lakes) can be understood and basing stability considerations on in situ density can represent thermobaric effects.

Clearly, the vertical dimension of the surface water and the deep water sections differs greatly between the numerical model and Lake Shikotsu. The inclusion of wind forcing at the surface as well as turbulent mixing in the upper water column in the model would result in a thicker surface water layer. Additionally, it would induce deeper mixing of cold water. Hence, the crossing with $T_{\mathrm{md}}$ would deepen and the deep water temperatures would decrease correspondingly. Additionally, a more sophisticated approach for diffusion based on the stratification strength of the water column could further refine the temperature

profiles, especially in summer. Due to strong summer stratification the heat transport into the deep water is presumably weaker compared to the model. Similarly, a limitation of the mixing range in each time step could further refine the evolution of the convective mixing cells. Also, other factors such as salinity would need to be included in a realistic simulation.

We tested the resilience of the thermobaric stratification against interannual variability. The simulations demonstrated the dependence of the deep water circulation on the surface temperature as driving force. They also proved that this type of circu-

lation was quite resilient against disturbances and returned to the usual circulation pattern within two or three years. Of course, changes in the governing influences of the circulation as mentioned above would modify the systems responses but the general behavior would be maintained.

## 5  Conclusions

In this paper, we presented an approach for implementing the effect of thermobaricity into a numerical model using in situ

density instead of potential density. We created a simplified 1D model excluding any external forcing except the surface temperature. Below the surface layer, dynamics were exclusively driven by diffusion and vertical mixing until stable density stratification was achieved. Moreover, we reduced the complexity to a minimum by omitting salinity, using horizontal homo-





geneity (one dimensional), a cylindrical bathymetry without shallow areas, and excluding any inflows. Hence, the thermobaric effects could be displayed without competing influences. To remain in a realistic range and have a good chance to reproduce
thermobaric effects, we used Lake Shikotsu as inspiration and used its maximum depth and surface temperature as input.

The model was able to conceptually characterize key features of the deep water circulation. It elucidated different phases of vertical mixing and defined the vertical and temporal extent of convection cells. Generally, the deep water circulation differed fundamentally from the conventional understanding (i.e. the mixing behavior when thermobaric effects are excluded). During winter stratification, cabbeling occurred at the intersection of the temperature profile with the line of the temperature of
maximum density $T_{\mathrm{md}}$ which induced convective mixing below due to thermobaricity. These mixing cells were disconnected from the surface and extended into a depth where the temperature was equal to $T_{\mathrm{md}}$ at the crossing or to the bottom. Hence, the depth of the crossing determined the resulting deep water temperature. This circulation behavior was not only influenced by the current surface temperature but also by the previous history of deep water renewal events. For instance, colder winter enhanced deep water circulation but led to weaker mixing in subsequent years.

In summary, this simplified model exhibits the necessity and the feasibility of including thermobaricity in the simulation of deep water circulation. This can be achieved by the implementation of in situ density (instead of potential density) for stability considerations. Despite the fact that this parsimonious approach could nicely reproduce the typical circulation features induced by thermobaricity, we are fully aware that a realistic representation of processes in Lake Shikotsu requires a proper lake model with complete forcing and accurate bathymetry. Hence, the proper solution for future lake modeling is the appropriate inclu-
sion of stability considerations based on in situ density into established lake models. This feasible approach will provide new insights into deep water formation in thermobarically startified lakes such as Lake Shikotsu and will improve the modeling of complex deep water renewal processes that are linked to thermobaricity as they occur in many large lakes.

*Code and data availability.*  The model code and input data used in this study is publicly available at GitHub: https://github.com/JMarks840/Thermobaric_Circulation.

*Author contributions.*  JM and BB conceptualized the study and designed the model approach. JM created the model, performed the simulations, analyzed and visualized the data, and wrote the original draft of the manuscript. BB acquired the funding, provided supervision and reviewed and edited the manuscript. KC helped with the logistics, acquiring the data and reviewed the manuscript.

*Competing interests.*  The authors declare that they have no conflict of interest.



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
