# Peer review of "Thermobaric circulation in a deep freshwater lake"

_EGUsphere, 2025_

## Author Comment (AC1)

Answer to:

RC1: 'Comment on egusphere-2025-1195', Anonymous Referee #1, 21 May 2025

Thanks for your review and your useful comments, we really appreciate this.
(The original comment in greyed out and italic and our response is black)

*This work is one of the very few attempts to understand and characterize the sequence of events leading to circulation—manifested as the cooling or warming of very deep water—in thermobaric, deep freshwater lakes using a simplified 1D model. The philosophy behind using this simplified 1D approach is to isolate the effects of thermobaricity and cabbeling, rather than focusing on wind-driven energy input or the complex hydrodynamics associated with realistic 2D or 3D bathymetry. That being said, the model successfully identified how the variation of the temperature of maximum density (Tmd) with depth under significant pressure alone (thermobaricity) can drive mixing in a deep lake.*

Thanks very much for the positive statements and briefly citing the focus of our work.

*The model was applied to a deep, cold Japanese caldera lake (Lake Shikotsu), where the hydrodynamics are believed to be predominantly vertical. It also demonstrates how using potential density at the surface may lead to completely different results compared to using stability criteria.*

We would not claim that we applied a model to Lake Shikotsu; we are clearly aware of the shortcomings for a realistic simulation (exchange processes at surface, lake basin, inflows / outflows, salinity). We rather used thermobarically stratified Lake Shikotsu as an inspiration and used few field observations to guarantee realistic boundary conditions.

- *The abstract would benefit from additional concluding sentences that elaborate on the key outcomes of the model, particularly the main physical features identified.*
  Our main conclusion is "Our results emphasize the feasibility and necessity of the implementation of thermobaricity in numerical lake models.", but we agree that we should include a sentence on reproducing thermobaric effects in a strictly one-dimensional model and its behavior (in addition to the smaller points that have been listed).

- *A clear distinction between thermobaric instability, thermobaricity, and cabbeling is needed, as these concepts are often confused. This clarification should be addressed consistently throughout the manuscript, including the conclusion. It would also be valuable to highlight that, in this case, cabbeling appears to result from eddy diffusion across Tmd at different depths—a particularly novel observation that, to my knowledge, has not been previously reported. As I understand it, this process involves the diffusion between a parcel of water already at a warmer temperature of maximum density (Tmd) and colder water, ultimately producing water at a lower Tmd. This mechanism deserves emphasis given its potential implications for deep mixing processes and how is it compared with "thermobaric instability".*
  We fully agree! In theory, thermobaricity and cabbeling are clearly separated: the previous covers effects deriving from the non-zero second order derivative of in-situ density after temperature and pressure, while the latter covers effects connected to the second order

derivation of in-situ-density twice after temperature. Despite this clear separation, cabbeling appears in our simulation of thermobaric effects and is recognized as the driving force for the deepwater convection. The resulting deepwater temperature (determined by the $T_{md}$ transition of the temperature profile) is a typical thermobaric feature. This means: we agree fully with the reviewer's observation; the separation can be difficult, but we promise to do our best to be clear in the new version of the manuscript.

- *There has been brief but noteworthy scientific debates regarding the appropriate criteria for evaluating stability, which merit mention. For instance, Georg Wüst (1932) and V. W. Ekman (1934) discussed the use of potential density—specifically, surface-referenced potential density—as a means of assessing stability. However, it is important to clarify that potential density referenced to an intermediate depth has since been recognized as a more reliable indicator.*

We trust that these old oceanographers had understood already 100 years ago how stability should be calculated: the additional compression of the deeper layer contributes to density but this does not add to stability. Even though the potential density at an intermediate depth is better than the potential density at the surface, it is still an approximation of using the in-situ density at every single point of the water column and still different to our approach. If stability is calculated at in-situ pressure then we achieve a good representation of thermobaric effects.

*This approach closely resembles what is being applied here, but at a common depth corresponding to the lower parcel,*

It is similar in so far, as the higher pressure effect on the deeper layer is removed by using the same pressure reference. However, our approach is fundamentally different, as our model uses the local pressure for stability calculations; hence, a different pressure for each stability calculation instead of one reference pressure for the entire model domain in space and time. Only with this approach a representation of thermobaric effects in a numerical model is done properly.

*and is supported by studies including Peeters et al. (1996), which also deserves mention. Finally, when considering which density measure to adopt, it may be useful to briefly reference the concept of quasi-density and explain why it has been excluded from the present analysis to contribute to the ongoing knowledge on the topic! It is very satisfying to see a comparison done with potential density at the surface, which I also believe one of the novel parts of this work.*

This "quasi density" of Peeters is a complicated quantity. We did not use this approach and hence we have not cited it. We will check again what it can be used for. However, we must avoid connecting "quasi density" to the simple conclusion of this paper: stability considerations based on in-situ density represent thermobaricity.

- *Why is the stability criterion being expressed in terms of density rather than simply using potential temperature, especially since salinity is excluded? (Gill, 1982; Imboden and Wüest, 1995). This approach might avoid the complications of selecting an appropriate density reference.*

We are fully aware of this shortcut using potential temperature (Boehrer PPNW contribution in Lake Tahoe, 2008). However, the goal of this paper is not the reproduction of a

temperature profile. The purpose of this manuscript is dealing with the theoretical side. We clearly prove that basing stability considerations on in-situ density covers thermobaricity. Building up on this, salinity can easily  be included in a next step, and a proper numerical lake model will be used.

*On that note, as mentioned in your manuscript (line 202), in-situ density is largely dominated by pressure, and there has been a brief scientific debate on the validity of using in-situ density for stability evaluation (A.H. Lee and G.K. Rodgers, 1972; Thomas Osborn and Paul LeBlond, 1974), ultimately ruling out its use. I believe what you are referring to in this publication is potential density at a common reference depth (at the lower parcel depth, not at the surface), which is conceptually like using an intermediate depth. It is not in-situ density, otherwise potential density at the surface is also in-situ density but the in-situ density at P2=0.*

In our understanding / convention, potential density refers to density at one reference pressure, which remains the same in the entire domain of the simulations and observations (in time and space, especially depth), while in-situ density represents the density at any given pressure (in-situ density can be calculated for depths other than the current location of the water parcel). We thought this is convention, but this comment tells us, we should explicitly write the definition out in the manuscript.

We would assume that for Lee, Rodgers, Osborn and LeBlond accounts the same as for above mentioned Wüst and Ekman: We may cite them in a general statement that they already have pondered how to evaluate stability from density profiles.

*An important consideration is what happens when this comparison crosses the Tmd line, as this transition is critical in our case: the compensation depth, which is defined relative to Tmd, governs the overall flow structure.*

"Compensation depth" is commonly understood as the depth where a displaced water parcel starts moving downwards as a consequence of its in-situ density compared to the in-situ density of its horizontal neighbouring water. This expression comes from the understanding of deepwater circulation being accomplished by displacing cold near surface water (by wind) in the vertical which is based on horizontal gradients. This is closely tied to the understanding of deepwater formation in Lake Baikal or similar cases. Our model is strictly one-dimensional and hence we exclude any horizontal gradients even in parametrized form. Therefore, and because in our one-dimensional model the deep circulation always starts at the intersection of the $T_{md}$ line and the temperature profile in contrast to the cases where the compensation depth is used we do not use the term compensation depth in our one-dimensional model.

*Also, I believe more justification is needed for the choice of evaluating density using the speed of sound (which is not measured, or maybe you have measurements not mentioned?), rather than the TEOS-10 approach utilizing potential temperature and salinity? As mentioned, it is mentioned that TEOS-10 "which includes the effect" compared to potential density, but still, potential density "at the surface".*

We used the sound speed because it is directly connected to the compressibility (sound velocity squared is equal the reverse ofthe compressibility times density). We will check whether this needs more explicit mentioning. In our formulation, the sound speed part represents the compression. Additionally, TEOS-10 is designed for ocean conditions. We believe that the pure water is better described by the formulas we use, since our model uses pure water.

- *I think it is worth defining the compensation depth. You later refer (line 71) to the intersection of the temperature profile with Tmd, which could be described as the compensation depth. It may help with clarity to introduce and use this term consistently throughout the manuscript.*

  As mentioned before, the "compensation depth" only makes sense when there is an environment to flow relative to. We think, it is generally difficult or even misleading to introduce expressions only to distance ourselves from them. We will check what might make sense.

- *It is mentioned that the temperature profile remains isothermal throughout. Is this monitored using thermistors or a CTD, and what is the measurement accuracy of this isothermal profile? For example, is the variability within 0.1 °C or 0.5 °C?*

  In the model, isothermal means isothermal = same number and convective cells shaded in gray in Fig. 4a have identical temperatures. In the measured profiles (Fig. 1), the homogeneous deepwater shows temperature variations in the vertical in the range of a few Millikelvin (this is visible from the thickness of the lines); in numbers +-0.002 Kelvin. However, this paper does not aim at a realistic representation of the situation in Lake Shikotsu: the gist of the paper is the conclusion that basing stability considerations on in-situ density represents thermobaric effects (already one-dimensional) and the approach is feasible and the effects are obvious and important for the circulation of deep lakes.

  *Clarifying this would help assess the significance of the observed isothermal conditions compared to the observed cooling/warming of the bottom water and also compare with other lakes. Additionally, where is the surface water temperature (model forcing) measured, and at what exact depth? In other lakes I believe it is usually at least 3-5 m deep in the surface mixed layer (I mean the shallowest thermistor).*

  The surface water temperature was measured roughly at 1.5 m depth (also subject to water table variation) at the end of the piers (we used existing structures in the protected national parc for placing the sensors). However, in winter, temperature differences in the surface water are very small. We will mention the sampling depth in the new version of the manuscript.

- *Can you provide a specific analysis or statistic isolating how much of the observed changes are driven by diffusion leading to "cabbeling" or "thermobaricity"? Additionally, how would changing the diffusion coefficient affect the overall process, since it seems like the main driver?*

  Right; our model aims at the representation of thermobaric effects. The diffusion implemented gives the vertical length scale. High diffusion results in a thicker surface layer. As we do not aim at a realistic representation of the situation in Lake Shikotsu, but at reproducing thermobaric effects in a numerical model, the vertical length scale is not essential. Also, the vertical length scale does not change the behavior of the described deep mixing in the model. In detail, the results show that in winter the vertical length scale is too small, i.e. diffusion in the model is much smaller than in Lake Shikotsu, while in summer, heat is forwarded too fast into the deep water and hence diffusion in the model during summer is assumed much higher than in Lake Shikotsu. However, this is fully disconnected from the

scope of our paper. This can be dealt with in future investigations.

*It is also unclear how the surface layer remains stable while convection occurs just beneath it that is (I believe) driven by cabbeling induced through diffusion? Clarifying this mechanism would help improve the physical interpretation.*
The overlying water is not included in the deep convection cell as its density is lower.

- *Why are some profiles perfectly following Tmd, and are they considered stable according to the used stability criterion? Because I would think that maybe again turbulent diffusion perturbations might deem these profiles unstable. That would be interesting to think about.*
  As long as temperatures are higher than $T_{md}(z)$, profiles are stable, because the expansion coefficient alpha is positive. In this range, turbulent diffusion does not contribute to instability, it rather stabilizes the overall picture.

- *I think you need to clarify more the particular use of ±0.4 °C for different climate scenarios, the selection of a three-year spin-up period, and the chosen value for the diffusion coefficient.*
  These simulations are not really climate scenarios. Instead, the different winter temperatures have been chosen to demonstrate that the intensity of the mixing in the deepwater depends on the winter conditions. The system returns into the typical stratification within few years. There is no justification for the +-0.4 Kelvin. Still, the simulation does not attempt to produce a realistic representation of the circulation in Lake Shikotsu.
  *Also, the method of mixing during the 1hour time step, is it sweeping downwards?*
  For each time step the whole water column is checked for stability bottom up. If two neighbouring layers are unstable they are mixed. Afterwards, this mixed part is compared with the neighbouring layers below the same way and so on until it is stable again. By this the whole water column is stabilized during each time step.
  *When does the algorithm stop?*
  As a consequence, the stratification is stable after each stability check. A repeated mixing is not required.

- *The discussion needs more comparison with previous studies especially with the closest model (Piccolroaz 1D model in 2013).*
  We will check what can be added.

*Specific notes:*

*Line 60: "Admittedly" I am confused from the structure of this sentence, what is being admitted?*
We will check this sentence.

*Line 76: Potential density "at the surface ". I think it is worth stating this whenever mentioned.*
We are not sure what the remark "at the surface" should indicate here,. This sentence in line 76 is correct since the pressure dependence gets lost by using the potential density no matter at which depth it is referenced.

*Line 105: So, this is the oscillation frequency using potential density at a common depth, not using in-situ density as it appears. Because in-situ means in its place, but you are evaluating both at P2, so it is*

*confusing. Using actual in-situ density gradient to evaluate N² would give a misleading sign as it is always dominated by pressure, hence again it is worth noting that this is not the in-situ density gradient, but the potential density or the density at a common reference that is the lower parcel depth.*

As mentioned before, the in-situ density can be calculated at different pressures as well. When using the (conventional) potential density only one or a few certain values for the pressure are used for the whole water column to get rid of pressure influences. However, we calculate the (in-situ) density for every pair of cells directly at their point of interaction, which we would consider in-situ. You are right, we use a common pressure for this comparison, but since this is different for every comparison it is the in-situ density and includes the compression of each water parcel even for the smallest movement to ensure correct stability considerations. That's why we stick to in-situ density to emphasize the inclusion of the compressibility in our calculations.

*Citation: https://doi.org/10.5194/egusphere-2025-1195-RC1*

---

## Author Comment (AC2)

Thanks a lot for your useful comments that help to improve our manuscript.
(The original comment is greyed out and italic and our response is black)

*Thermobaric circulation in deep freshwater lake by Marks et. al.*

*In this work, the authors undertake a numerical process study to demonstrate the effects of a thermobaric circulation in a cylindrical domain. This domain is inspired by Lake Shikoku, which has been previously observed to undergo thermobaric circulation. The authors employ a 1D column model to explore the vertical transport of heat over several simulated years. The main crux of the argument is that by considering the in-situ density (as opposed to the potential density which excludes thermobaric effects a priori), the authors identify the process by which thermobaric effects effectively mix the water column. Overall, I thought this article was put together well, and interesting. I have a few concerns, however, that should be addressed prior to publication.*

Thanks for the short summary of our manuscript and the positive feedback of the topic.

*Main points*

- *I am certainly empathetic to the process study approach utilized by the authors, and I'm happy to read work that uses an idealized approach to learn about different process in isolation. I am, however, wondering about the relative importance of thermobaric effects to other, potentially more vigorous, dynamical effects, especially those ignored in this study. I think a discussion on this topic by the authors would help the framing of the work.*

  The reviewer is right, other effects like fluxes at the surface are as important for the circulation in the lake at least at some depths. The parameterization of surface fluxes has been the extensive investigation of recent years and decades. However, the possible dominance of e.g. surface fluxes requires their exclusion from the demonstration of thermobaric effects, otherwise 1) thermobaric effects could be mis-interpreted as possible secondary effects of other driving forces and 2) the effects of thermobaricity could not be clearly demonstrated when other effects of similar importance are interfering. We are thankful for the advise and we will check, whether the arguments need clearer statements in the text.

- *Related to the above point, on line 111, the authors comment "this kind of deep water renewal is suspected to have a significant influence in this lake", and I'm wondering if they could clarify if they think this based on the observational data, or for some other reason, such as the depth.*

  Right, the statement is based on observational data: the homogeneous water below 4 °C in the deep layers of Lake Shikotsu, and the intersection of the $T_{md}$ line defining this temperature. Probably we should add "see Fig. 1b)" in the text to make this clear.

- *While I was reading this work, I kept asking myself "What is the specific thing that thermobaricity is doing that's different"? It wasn't until I read section 3.5 that things (sort of) cleared up for me, though I'm still not quite sure.*
  - *In my opinion, the argument the authors try to make could be strengthened by first using section 3.5 as a straw man, and then discussing the new results (i.e. the results WITH thermobaricity). I think the authors even have their conclusions laid out this way already. Related to this, I encourage the authors to add a picture similar to figure 4, but for the "non-thermobaric" case. I think that would strengthen their argument for "what thermobaricity does".*

Thanks for the suggestion of restructuring our manuscript in favor of better clarification. We will try to optimize the structure and also emphysize the new features we have found. Regarding a similar figure to figure 4, this would be figure 6. If you meant figure 3 we will consider this input in the revised version.

  - *I sort of understand what the authors are getting at in the "Convective Mixing" section, but I think some sort of schematic explaining the convective cell detached from the surface looks like, or maybe an arrow placed on figure 4(b) describing what they mean. (This would certainly aid in my understanding).*

Thanks for pointing out this issue. We will try to clarify this in the manuscript.

***Minor Points***

*There are typos in a few places (eg lines 73, 77, 112, 118, and a few more). Please carefully check the manuscript*

Thanks for pointing out, we will check that.

*Line 36: Can the authors clarify what they mean? This sentence beginning with "Ultimately..." is confusing and I'm not sure what the authors mean.*

We wanted to state: At the surface, water below 3.98°C is more dense than slightly colder water, but from a certain depth, this density difference reverses. We will reformulate this.

*Can the authors provide a little more info on how they arrived at equation (6). It's not so clear to me, but I think they've taken the derivative of rho_pot (rearranged from equation (3)), and then made the approximation that rho_pot \approximation rho_in-situ in the denominator of equation (6)?*

In principal that connection can be made, but we did not used this derivation directly. We will clarify the derivation in the revised manuscript.

*The authors mention that p=0 corresponds to atmospheric pressure at the surface (line 140), but this convention is employed (equation (4)) before it is mentioned in (section 3). Please mention this convention upon first usage.*

We will correct this.

*Lines 94-97: it's not clear to me what you're saying here. Is this maximal deviation the deviation that occurs over 360 m, or between 3 and 4 deg(C), or something else? The sentence in line 96 seems to imply it's something else.*

It is actually the maximum deviation between the linear approach (Eq. 4) and the tabled more exact values. We will reformulate this sentence.

*Line 133: Can the authors clarify what they mean by this sentence? I'm getting confused by the use of the words in the parentheses.*

Mixing layers receive the average temperature, except if mixing includes the surface layer; then the temperature of all mixing layers is set to the surface temperature. (We will clarify this in the manuscript.

*Line 203: "Stability frequency" is used. Is this standard? "Brunt-Väisälä frequency" is used in the abstract. I would standardize the usage throughout the paper.*

Yes, we will standardize this in the manuscript.

*In figure 1, pressure on the vertical axis is positive, but on the subsequent figures, it's negative. I would suggest that it be standardized to one or the other, or clarified in the text.*

Thanks for pointing out this. We will correct figure 3 so that every figure has the same (positive downwards) pressure y-axis.

*The authors model convection in a phenomenological way (i.e. all heat is exchanged between adjacent layers instantaneously). For the purposes of this work, I think it's probably fine, but I don't really know. Can the authors comment on whether they think that their approach is actually a good representation of the true convective processes going on in a lake? I.e. are the timescales appropriate? Is there evidence of a lake-wide circulation?*

As you correctly mentioned we only aimed for a phenomenological representation of the lake mixing. Hence, our time scales regarding the mixing are not aimed to be realistic in the short term. During the year we think that the mixing patterns are represented conceptually correctly, if compared to the measurements of Lake Shikotsu. To get a more realistic time scaling we would need to implement more features into the model that would complicate it. We think that the phenomenon of the mixing is different to reality with regard to the exact length in time for the different phases, but not in the form of the conceptual phases themselves. The lake wide circulation can be assumed because of the measured profiles during the mixing phase in Lake Shikotsu.

*Citation: https://doi.org/10.5194/egusphere-2025-1195-RC2*

---

## Author Comment (AC3)

Answer to:

RC3: 'Comment on egusphere-2025-1195', Anonymous Referee #1, 03 Jun 2025

Many thanks for your input. It will surely improve the manuscript.
(The original comment in greyed out and italic and our response is black)

*This manuscript uses a vertical 1D idealized model to capture thermobaric effects on the seasonal evolution of thermal stratification. The aim of the manuscript is to highlight physical processes the dominate the seasonal cycle. The model implements a novel estimate for gravitational stability, as well as simplified vertical diffusion and surface thermal forcing. Using these three simple concepts they are able to reproduce the basic characteristics of observed thermal stratification in Lake Shikotsu, Japan, a caldera lake whose thermal dynamics are believed to also be mostly vertical 1D.*

Thanks for the short summary of our work.

*Much is made of the novel implementation of gravitational stability, but little is said of what previous modellers have done, eg Killworth et al (1996) and Piccolroaz and Toffolon (2013). How is the formulation derived here different and what are its advantages over other, existing formulations?*

For pure water in the limnic temperature range, the appropriate density formulation is either Tanaka et al. (2001) or Kell (1975). The compressibility of pure water in the limnic range is most accurately concluded from sound speed tabled by Belogolskii et al. (1999). The special feature of Chen and Millero is the inclusion of salinity in their density formulation. However they base the salinity on ocean composition and as a consequence the density contribution of salts is badly underestimated for most freshwater lakes (see Moreira et al. (2016)). Hence Chen and Millero is not the appropriate approach for our model set-up. We use the proper approach for pure water.

*The distinction between instability induced by vertical displacement of a stable profile to a depth where it becomes unstable ("forced plume downwelling") and mixing of waters ("cabbelling") is an important one. While it shows up in the introduction (Line 40-50), it gets a bit lost in the rest of the text. For example, line 50 seems to equate cabelling with simply "thermobaricity". I recommend clearly and consistently delineating and labelling these two processes throughout the ms. To me, the most interesting result of this work is the focus on how surface convection interacts with the Tmd line subject to cyclical surface forcing.*

 You are right, we will improve the distinction between the different instabilities to clarify the processes identified in our model. Regarding the cabbeling, we will clarify that this is the initiating process that induces the downwelling at the intersection of the $T_{md}$ line with the temperature profile, and separates the deep downwelling from the surface circulation, while thermobaricity sustains the downwelling into the deep water.

*I encourage the authors to say more about the surface forcing. You use an hourly timestep to resolve the diurnal evolution. Cite or specify some details about the surface measurements (eg depth, sampling interval, instrument details). Why was it important to resolve the diurnal cycle? Do the results change if you use daily averages?*

We will add the depth of the surface water temperature sampling. We resolved the diurnal cycle to include the lower temperatures at night and the higher temperature during the day.

*I would like the authors to say more about the two mixing processes built into the model (convective readjustment and diffusion) and how they interact. Currently the manuscript focuses on calculation of stability and subsequent convective readjustment, but says little about the effects of what appears to effectively be a background constant diffusivity set to a rather high value, especially for the deep waters of a deep lake with relatively small surface area. How sensitive is the model to the chosen value of diffusivity (or (time step)/(grid size) ratio)? How does the diffusivity interact with convective instability correction? Why did you even add diffusivity? Presumably the results are very different without it.*

The diffusivity provides the vertical length scale of processes. Smaller values would still provide a similar picture but on a smaller vertical scale (and hence also smaller temperature differences as a consequence of thermobaricity). We used a reasonable value and conclude that at some locations (times) it is too small and at other locations (times) it is too large. However, we refrained from using varying values (in time or space) to exclude the possibility of circulation or resulting temperature profiles possibly being created by this. Diffusivity was added, because otherwise the entire temperature profile would be pulled onto 4 °C and remain there, except the surface layer following the implemented surface temperature.

*I find the use of "in situ density" to describe the stability model to be misleading. The formulation for stability developed in this ms seems to be a discrete approximation of potential density using a reference pressure at the lower of the two grid cells being compared. Put another way rho(T1,p2) can be said to be potential density from cell one evaluated at a reference pressure of cell two. I would be more comfortable saying either stability was estimated by "accounting for compressibility effects using local temperature and pressure", or "using potential density with local reference pressure", or something like that. I appreciate that the authors have written the formulation of stability in terms of density and drho/dp (or c or 1/bulk modulus) rather than temperature and alpha (thermal expansion coefficient), and there isn't really a word for "compressibility effects" in this novel density formulation in the same way there is a thermal expansion coefficient (ie alpha) for a temperature formulation.*

We are absolutely consistent in our density convention: potential density is $rho(T,p_0)$ where $p_0$ represents a pressure reference, which is kept constant over the model domain in space and time. In-situ density is $rho_{IS}(T,p)$ at any pressure p. Maybe we should write this out somewhere, but we thought this is textbook knowledge.

*Minor/editorial comments*

*Title: should include words like model and 1D.*

We did not use the word model in our title so far because we do not want to give the impression of creating a realistic model for a lake but rather give a conceptual impression of the thermobaricity driven deep water circulation. But we will think about optimizing our title.

*Abstract: The abstract includes a lot of introductory and methodology, but no results. This reads more like an aspirational conference abstract, rather than a complete work published in a journal.*

Our key outcome is mentioned in the last sentence but we will check what other results to add in more detail in the abstract.

*Line 29: "The effect deriving from this property is called thermobaric effect". This sentence is not very helpful in defining what you mean by "thermobaric effect" or "thermobaricity". This is a good place to clearly define it, especially if you plan to use it to differentiate from "cabbeling" (Lines 39-44) or "forced plume downwelling" (line 50)*

Thermobaricity is explained shortly after (line 32and following) it was mentioned first, but we will think about optimizing the sentence.

*Line 40: "Cabbeling originates from thermal bars…" seems misleading and not very helpful. One might also say thermal bars originate from cabbeling. I recommend simply saying "Cabbeling occurs where …"*

We will check this.

*Line 43: "Although deep water renewal in some lakes is controlled only by thermobaricity, also cabbeling may be involved in the deep mixing…". Without a definition of thermobaricity it is not clear what you mean by "some lakes". Which lakes? What are their properties? Give an example of one that is controlled only by thermobaricity and not cabbeling.*

We will try to add information here in the revised version of the manuscript.

*Line 47: Define "compensation depth". Also, "proceed" where?*

We will clarify this.

*Line 44: state explicitly the "convenient property of potential density" you are referring to.*

We will check this, see also above.

*Line 49 and 50: These two sentences together are very confusing. You are contrasting deep water mixing from wind forced downwelling under conditions of thermobaricity (ie "forced plume downwelling") with "thermobaricity". What is the difference? How are these not both "thermobaric effects"?*

You are right, both are thermobaric effects. Our formulation here is not optimal and we will change this.

*Line 50: Who are "them"*

"Them" are the previous mentioned models of Killworth et al. (1996) and Piccolroaz and Toffolon (2013). We will change this in the manuscript.

*Line 67: "temperatures" should be "water temperatures"*

We will change this.

*Line 75: "my" should be "by"*

We will change this.

*Line 90: Tell us why it is ok to ignore the effects of local limnic chemistry that "must be included"*

We ignore salinity in our considerations (lines 90 / 91)*Eqn 6: Highlight in the text that rho_1 is evaluated at p_2. This is key to the whole scheme and could easily be missed by the reader. This might also be a good place to say something about rho_(in situ) evaluated at p_2 isn't really "in situ" anymore, but effectively potential density using a grid specific reference pressure.*

We will check the description. Of course, it is possible to calculate in-situ density of a water parcel without taking it there (see definition of in situ density).

*Line 118: "May" is misspelt*

We will change this.

*Line 125-135: More information about the numerical scheme is warranted to help understand the results. What is the order of operations? From the text it looks like the surface boundary condition is updated first, then diffusion occurs, then stability is considered. If this is the order, say so. Are the diffusion and stability calculations done in an upward or downward sweep? Also, this would be a good place to explain why diffusion is needed in the model. What are the implications of neglecting it? How sensitive is the model to time step and layer thickness, which controls the effective diffusion, e.g. why is half the volume exchanged each hour?*

The order you mentioned is correct, we will add this. The diffusion is done simultaneously for the whole water column, while the stability is checked bottom up. For the latter the unstable layers are mixed downwards again so that only one stability check generates a stable water column. The influence of the diffusivity is explained above. We will check how to improve the manuscript here.

*Line 170: It is worth pointing out that "summer warming" occurs over 25 hours.*

We do not aim for a realistic model (neither in time nor space) and we do not use a realistic representation of the diffusivity in time and space (compare above). Therefore, we would refrain from pointing this out since it has neither a specific implementation in the circulation characteristics nor a realistic basis.

*Line 173: "WS2" I think should be "SW2"*

We will correct this.

*Line 188: Who are "They"?*

The small rewarming events. We will improve this.

*Line 195: I think you mean "SW3 and SW4" here*

We will correct this.

*Line 225: If breakdown of prior strong summer stratification is important, then results will be sensitive to the linear interpolation of summer temperatures from May through October. In particular the summer peak will be missed. Would it make a difference if you interpolated linear to an estimated summer peak surface water temperature?*

It would change the strength of the summer warming and could, regarding the strength, influence the following winter circulation as explained in the text. However, since we do not have a realistic representation of a lake with this model and only aim to conceptually show the influences of the

different surface temperatures the usage of the linear interpolation is sufficient in this case. Using the summer peak (which is not included in our input data) would not change the in the manuscript described behavior.

*Line 225: "Strong winter period" is unclear*

We mean a colder winter. We will clarify this.

*Line 233: I don't understand what you mean by "every transition of maximum rho_pot"*

This is every time the surface temperature crosses the temperature of maximum density at the surface, about 3.98 °C, where the potential density is at its maximum.

*Line 248: "similar lakes" Similar how?*

Deep lakes with deep water temperatures below 3.98 °C during summer stratification.

*Line 266: What is the difference between "diffusion and vertical mixing"?*

We mean the vertical mixing induced by instability. We will clarify this in the manuscript.

*Line 277: "the depth of the crossing" is unclear*

The crossing of the temperature profile and the $T_{md}$ line. We will clarify this in the manuscript.

***Citation**: https://doi.org/10.5194/egusphere-2025-1195-RC3*